# Distributionally Robust Optimization as a Scalable Framework to Characterize Extreme Value Distributions

**Patrick Kuiper**[*1]   **Ali Hasan**[*1]   **Wenhao Yang**[2]   **Yuting Ng**[1]   **Hoda Bidkhori**[3]   **Jose Blanchet**[2]   **Vahid Tarokh**[1]

[1]Dept. of Electrical and Computer Engineering, Duke University, Durham, North Carolina, USA
[2]Dept. of Management Science and Engineering, Stanford University, Stanford, California, USA
[3]Computational and Data Sciences Dept., George Mason University, Fairfax, Virginia, USA

## Abstract

The goal of this paper is to develop distributionally robust optimization (DRO) estimators, specifically for multidimensional Extreme Value Theory (EVT) statistics. EVT supports using semi-parametric models called max-stable distributions built from spatial Poisson point processes. While powerful, these models are only asymptotically valid for large samples. However, since extreme data is by definition scarce, the potential for model misspecification error is inherent to these applications, thus DRO estimators are natural. In order to mitigate over-conservative estimates while enhancing out-of-sample performance, we study DRO estimators informed by semi-parametric max-stable constraints in the space of point processes. We study both tractable convex formulations for some problems of interest (e.g. CVaR) and more general neural network based estimators. Both approaches are validated using synthetically generated data, recovering prescribed characteristics, and verifying the efficacy of the proposed techniques. Additionally, the proposed method is applied to a real data set of financial returns for comparison to a previous analysis. We established the proposed model as a novel formulation in the multivariate EVT domain, and innovative with respect to performance when compared to relevant alternate proposals.

## 1  INTRODUCTION

Modeling rare and extreme events is an important task in many disciplines such as finance, climate science, and medicine [Dey and Yan, 2016]. Estimating distributions of rare events from data is difficult due to the lack of observations within this region, making it challenging to understand the risks deep in the tail of a distribution. Extreme Value Theory (EVT) studies the class of distributions arising as the possible distributional limits that can be used to estimate multivariate distributions in distant (relative to the origin) regions (i.e., tails) which by their nature witness very few observations (or none at all). These distributional limits are derived as the possible asymptotic statistical laws of shifted and re-scaled data as the sample size increases. It turns out that such possible distributional limits form a semi-parametric class called *max-stable distributions*, which are constructed in terms of a spatial Poisson point process.

Naturally, because of the lack of data in extremal regions and because of the asymptotic nature of max-stable models, their use in inferential tasks involving tails is exposed to high variance due to model misspecification. Moreover, when using max-stable models, the assumptions that the data converges to the distributions specified by EVT must be made. This leads to an important question: *How can we robustify against scenarios deep in the tails while observing potentially sub-asymptotic data where the assumptions of EVT may be violated?* To answer this question, we propose a solution based on distributionally robust optimization (DRO). DRO involves a zero-sum game in which the statisticians play against an adversary that perturbs (in a non-parametric way) the nominal / baseline distribution assumed by the statistician. Building on classical DRO, we carefully design constraints to retain the extrapolation properties of EVT for the robustified distribution.

Since we are interested in robustifying the tails, we will consider max-stable baseline distributions given by extreme value distributions (EVDs), as presented by de Haan and Ferreira [2010]. *Max-stability* roughly states that the distribution of the maximum of independently and identically distributed (i.i.d.) samples belongs to the same distribution up to a change in the location and scale parameters. This means that the "shape" of the distribution is preserved under the max operation, and it is this property that allows for extrapolating to regions outside the observation domain. In our robustification framework, we therefore wish to preserve

---

*Authors equally contributed to this work

this max-stability property. We do this by searching over the space of distributions that are also max-stable, in effect constraining our search to only distributions that extrapolate according to EVT, which form a semi-parametric class. We do this by carefully designing the uncertainty set such that the necessary max-stable properties are preserved.

To illustrate our desired result, we refer to Figure 1. This figure visualizes how a completely unconstrained adversary (solid line) may be too conservative and may not consider the extrapolating properties of the distribution. A well constrained adversary to a max-stable distribution (dashed line) provides an appropriate balance of coverage while maintaining underlying structural properties. This figure demonstrates a case when even if we consider two adversarial formulations that achieve a similar minimum error value, the properly constrained adversary (dashed line) is preferable because the size of uncertainty is very hard to calibrate. Therefore, a curve that is "flatter" around the minimum as a function of the uncertainty size parameter is preferable.

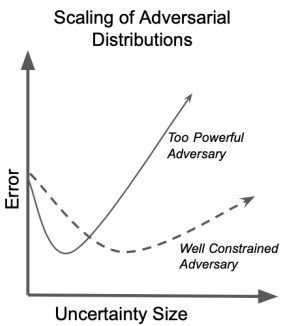

Figure 1: Illustration of the error in expected loss with two models under different constraints. Here uncertainty size describes a confidence in the data used for extrapolating EVT distributions, usually quantified by the amount of data available for this process

Our work focuses on the case of multivariate extreme value distributions (MEVs) which characterize the joint risk between different variables. Unlike univariate EVDs which have a fully parametric form, MEVs are semi-parametric, where the dependence structure is an infinite dimensional object, and much harder to estimate. We focus on robustifying the dependence structure while preserving the MEV character. This representation leads to additional difficulties which we overcome through the proposed DRO framework.

In view of the fact that the lack of data will naturally induce model error, we introduce an approach to quantify model misspecification based on optimal transport DRO. We select the Wasserstein metric for optimal transport because this approach, together with moment constraints, encompasses most DRO formulations as demonstrated by Blanchet et al. [2023].

The cost structure in the optimal transport discrepancy allows one to balance various objectives when modeling data, specifically the trade-off between tractability and control of pessimism in extremal behavior. A too-powerful adversary makes the size of uncertainty hard to calibrate since a slight increase may result in adversarial policies that perturb a distribution in ways that are too pessimistic and may not be consistent with the constraints imposed by EVT.

We mitigate these concerns by studying DRO formulations which constrain the adversarial perturbations to explore non-parametric models that induce robust estimators while preserving MEV characteristics. One of the approaches we employ is based on optimal transport for point processes, while the other is based on a neural network architecture. We illustrate the performance of the neural network architecture in the context of a Conditional Value at Risk (CVaR) metric applied to a multi-variable extreme value distributed data set. The network is evaluated across several synthetic data sets, specifically constructed to challenge the assumptions associated with EVT. The neural network architecture uses the tractability of the optimal transport distance metric to parameterize model uncertainty. Furthermore, our analysis is extended to a real data set of financial returns as a baseline comparison, similar to the data proposed by Yuen et al. [2020].This experiment demonstrates precisely the anticipated behavior shown in Figure 1, where our methodology leverages a properly constrained adversarial estimator.

**Related Work** A number of related research directions exist that consider both estimating EVDs from data as well as robustifying them using DRO. Since our focus is on multivariate EVDs, we will review a few of the estimators from the literature. For estimating multivariate EVDs, copula based approaches have been developed in a variety of instances including work by [Gudendorf and Segers, 2010, Marcon et al., 2017, Hasan et al., 2022]. Samplers for MEVs have also been considered as demonstrated by [Dombry et al., 2016, Liu et al., 2016]. Hasan et al. [2022] provides a flexible framework that uses neural networks to estimate and sample from multivariate EVDs irrespective of dimension, which we use in the computational component of this work. However, all of these methods only consider the case where the model is well-specified and do not consider uncertainty associated with the model class. We build upon these works with the addition of the DRO perspective.

With regards to the DRO literature, Blanchet and Murthy [2019] described the general framework for Wasserstein DRO that we will use throughout this work. Additional discussion is available in [Rahimian and Mehrotra, 2022] and [Van Parys et al., 2021]. Other DRO frameworks for EVT have been considered, but only in the univariate case, by [Blanchet et al., 2020, Bai et al., 2023]. Blanchet et al. [2020] considers DRO under the Kullback-Liebler divergence, which is appropriate for the univariate analysis but may not be appropriate for the multivariate case where sup-

ports are likely to be disjoint.

Finally, Yuen et al. [2020] considers DRO estimators specifically for MEV distributions leveraging extremal coefficient constraints. In this work, upper and lower bounds are established on a Value at Risk (VaR) loss and applied to a real data set of financial returns. These bounds are established over the infinite domain of spectral measures, using a finite set of constraints formulated as a linear semi-infinite program. This is a limited subset of application problems when compared to our investigation. While the general goal of Yuen et al. [2020] is similar to our framework, our method extends to more general risk measures and considers a flexible uncertainty set specified by the Wasserstein distance. Furthermore, we achieve a single robust loss (upper bound), as opposed to less precise (upper and lower) boundaries. In Section 5 we generate a similar data set to [Yuen et al., 2020] and employ our proposed method for comparison. In summary, the main contributions of this paper are:

- We introduce a framework to produce DRO estimators for MEV distributions based on optimal transport.

- We provide tractable DRO formulations for various estimators of interest when adversaries live in the (infinite dimensional) space of point processes.

- We test the performance of our estimators with the goal of showing that our MEV-constrained adversaries improve performance in the sense of Figure 1, and compare to a previous work for baseline analysis.

## 2 BACKGROUND AND PROBLEM FORMULATION

In this section we focus our discussion on concepts critical to our proposed results and we introduce the framework for distributionally robust estimators in MEV distributions. We first provide a brief overview of EVT and describe how it is used to extrapolate beyond the observed data. We then introduce the multivariate counterpart, which we use throughout this work, to describe EVT in multi-dimensional settings. Finally, we discuss the distributionally robust optimization framework we use based on the Wasserstein distance.

### 2.1 EXTREME VALUE THEORY BACKGROUND

We begin by reviewing concepts surrounding MEV distributions. Consider a sequence of $n$ i.i.d. random vectors $\{X^{(1)}, \ldots, X^{(n)}\}$, with $X^{(i)} \in \mathbb{R}^d$ and $i = 1, \ldots, n$ and denote the maxima over each dimension as $M_{k,n} := \max_{i=1}^{n} X_k^{(i)}$, where $k \in \{1, \ldots, d\}$. Similarly to univariate EVT analysis, we consider MEV distributions $G$, where $P((M_{1,n} - b_{1,n})/a_{1,n} \leq z_1, \ldots, (M_{d,n} - b_{d,n})/a_{d,n} \leq z_d) \rightarrow G(z_1, \ldots, z_d)$, for some normalizing constants $a_{k,n} > 0$ and $b_{k,n}$, as the number of observations $n$ increases to infinity.

### 2.2 SPECTRAL REPRESENTATION OF COMPONENT-WISE MAXIMA AND ASYMPTOTIC CHARACTERIZATION

We will now introduce specific properties of MEV distributions that we will exploit in our framework. Consider i.i.d. random vectors $\{X^{(1)}, \ldots, X^{(n)}\}$, with $X^{(i)} \in \mathbb{R}^d$ and $i = 1, \ldots, n$, with unit Frechét margins such that $F_{X_k}(x) = \exp(-1/x)$, with $x > 0$ for all $k = 1, \ldots, d$. Following Coles [2001], let a sequence $N_1, \ldots, N_n$ be a point process, where $N_n(\cdot) = \sum_{i=1}^{n} \mathbb{1}_{\frac{X^{(i)}}{n}}(\cdot)$ with $N_n(\cdot) \xrightarrow{d} N(\cdot)$ as $n \rightarrow \infty$ with $d$ denoting convergence in distribution and $N$ is a Poisson point process. We will apply this result further in Section 3 to define the proposed robustificaiton of the Poisson point process.

**Decomposing Max-Stable Random Variables** Max-stable distributions can be decomposed according to the radial and spectral decomposition [Dombry et al., 2016, Liu et al., 2016]. Specifically, if we let $Y^{(n)} \in \mathbb{R}^d \sim H$ be a sample from the spectral distribution and $A^{(n)}$ to be the $n^{\text{th}}$ arrival of a unit rate Poisson point process, then a max-stable random variable is represented as

$$M = \max_{n \geq 1} \frac{Y^{(n)}}{A^{(n)}}. \tag{1}$$

Under the condition that $\mathbb{E}[Y_k] = 1$, the variable $M$ is distributed with unit Frechét margins. This decomposition provides a semi-parametric class of distributions whose structure we will use throughout the rest of the text. With the spectral decomposition of the MEV we are now able to analyze an MEV distribution. Additionally, we introduce Lemma 2.1 which allows for the explicit transformation of the MEV cumulative distribution function (CDF).

**Lemma 2.1** (A corollary of Theorem 1 in de Haan [1984])**.** *For $X^{(k)}$ with standard Frechét marginal distributions, we have:*

$$P\left(\frac{M_{1,n}}{n} \leq x_1, \cdots, \frac{M_{d,n}}{n} \leq x_d\right)$$
$$\rightarrow \exp\left(-V(x_1, \ldots, x_d)\right),$$

*where*

$$V(x_1, \cdots, x_d) = d \int_{\Delta_{d-1}} \max_{k=1}^{d} \frac{w_k}{x_k} H(dw),$$

$\Delta_{d-1}$ *represents the unit $d-$dimensional simplex and $H(\cdot)$ satisfies: $\int w_k H(dw) = d^{-1}$ for all $k \in \{1, \ldots, d\}$.*

When dealing with MEVs, the important component in estimation is the spectral measure $H$. When placed in its standardized form, $H$ is a measure with support over the simplex that describes the dependence between the covariates, specifically which components become extreme simultaneously.

When a face of the simplex has $H$ non-zero, it implies that covariates associated with the vertices of the face experience extreme events simultaneously. For more details on estimating the spectral measure, we refer the reader to [de Haan and Ferreira, 2010, Gudendorf and Segers, 2010]. In Section 3, we describe how $H$ plays the role of an intensity function when taken in the perspective of the point process.

## 2.3 WASSERSTEIN DRO FORMULATIONS FOR EVDS

Referencing the framework to quantify model uncertainty via DRO described by Blanchet et al. [2020], we define the probability space $(\Omega, \mathcal{F}, P)$, where a candidate robust distribution $P$ is feasible such that $d(P, P_0) \leq \delta$. $P_0$ is a baseline distribution and $d(\cdot)$ is a distance measure, constrained by the parameter $\delta$. Two methods are commonly employed to quantify model uncertainty when constructing distributional ambiguity sets. The first considers the corruption of the likelihood baseline model to be misspecified, which is addressed via $\phi$-divergence ambiguity sets. The second method perturbs the actual data, which leads to the use of Wasserstein distances to quantify model misspecification.

A recent investigation by Blanchet et al. [2023] has demonstrated that both considerations of likelihood and perturbations of data may be unified under the Wasserstein distance, where $d(\cdot) = W_c(\cdot)$. Consider a loss function $\ell : \mathbb{R}^d \to \mathbb{R}_+$ and the Wasserstein distance transport cost $c : \mathbb{R}^d \times \mathbb{R}^d \to \mathbb{R}_+$. We define the primal optimization problem as follows:

$$\max_{P : W_c(P, P_0) \leq \delta} \mathbb{E}_{X \sim P}[\ell(X)] \qquad (2)$$

In the context of our problem, $P_0$ could be the estimated distribution from available samples, but one may not have enough coverage since rare events may not have been observed within the data collection period. Following Blanchet and Murthy [2019], the dual form of (2) is given by the following problem

$$\min_{\lambda \geq 0} \left[ \lambda \delta + \mathbb{E}_{X \sim P_0} \left[ \max_Z \ell(Z) - \lambda c(X, Z) \right] \right]. \qquad (3)$$

As we will discuss, the dual form is particularly amenable for computational purposes, especially when one can exploit specific properties of $c$, $\ell$, and $P$. Our investigation will describe two perspectives of (3) for EVDs. The first we present in Section 3, and is based on interpreting an MEV through a point process and the second is based on the interpretation using a copula. Each perspective has different properties that are useful for computational purposes, which we will discuss in Section 4.1.

## 3 ROBUSTIFICATION OF THE POISSON POINT PROCESS

In this section, we provide a more general formulation for DRO problems using a characterization with a point process. For the probability space $(\Omega, \mathcal{F}, P)$, we denote a counting measure $N(\cdot)$ on a Polish space $(S, d)$. We define $N(\cdot) = \sum_{i=1}^{\infty} \mathbb{1}_{x_i}(\cdot)$ and $N'(\cdot) = \sum_{i=1}^{\infty} \mathbb{1}_{y_i}(\cdot)$ as two random counting measures. Additionally, we define scaling functional $\kappa : \mathbb{Z} \to \mathbb{R}_{\geq 0}$, and a lower semi-continuous cost function $c : S \times S \to \mathbb{R}_{\geq 0}$. Let $\sigma(\cdot)$ denote the permutation function defined on the support of $N(\cdot)$, and we then define the distance as:

$$\widetilde{c}(N(\cdot), N'(\cdot)) = \infty \mathbb{1}_{N(S) \neq N'(S) \text{ or } N(S) \vee N'(S) = \infty}$$
$$+ \kappa(N(S)) \inf_{\sigma(\cdot)} \sum_{i=1}^{N(S)} c(x_i, y_{\sigma(i)}). \qquad (4)$$

This distance is also similar to the ones presented by Barbour and Brown [1992], Chen and Xia [2004], Gao and Kleywegt [2023]. Additionally, we define the 1-Wasserstein distance between two random counting measure by:

$$W_c(\mu, \nu) = \inf_{\gamma \in \Gamma(\mu, \nu)} \mathbb{E}_{(N, N') \sim \gamma} \widetilde{c}(N, N'), \qquad (5)$$

Here $\mu$ is the measure corresponding to $N(\cdot)$ while $\nu$ is the measure corresponding to $N'(\cdot)$. Analogously to standard DRO problems, we can write down a new formulation for this DRO problem with point process as follows:

**Theorem 3.1** (Primal and Dual Form for Point Processes).
*Let $N(\cdot)$ denote a Poisson point process, $f(\cdot)$ denote a Borel function, and $W_c(P, P_0)$ denote the Wasserstein distance between measures induced by $N'$ and $N$ under cost metric c. For a distance $\delta \geq 0$, the following problems are equivalent:*

$$\sup_{W_c(P, P_0) \leq \delta} \mathbb{E}_{N' \sim P}\left[\ell(N'(f))\right] =$$

$$\inf_{\lambda \geq 0} \left\{ \lambda \delta + \mathbb{E}_{N \sim P_0} \left[ \sup_{N'} \left[ \ell(N'(f)) - \lambda \widetilde{c}(N, N') \right] \right] \right\},$$

*where $N(f) := \int f(x) N(dx)$.*

*Proof.* This follows from the application of Blanchet and Murthy [2019]. $\qquad \square$

As mentioned in the background material in (1), MEV observations are given by the product of radial $(A^{(n)})$ and spectral $(Y^{(n)})$ components. In the perspective of the point process, the associated measures for each of these components forms the intensity of the point process. Specifically, we let the atoms of the point process be given by $(A^{(n)}, Y^{(n)})$ where $A^{(n)}$ can be thought of as a time coordinate (since $A^{(n)}$ is the $n^{\text{th}}$ arrival time) and $Y^{(n)}$ as a space (or the mark) component. Define $N(da, dv) = \sum_{n=1}^{\infty} \mathbb{1}_{((A^{(n)}, Y^{(n)}))}(da, dv)$.

This provides a useful characterization in terms of the dual problem insofar as we can optimize over the *boundaries* of the integrals to form our adversary point process. As we will illustrate, this is equivalent to finding a new point process with modified intensity.

With regards to the cost in the Wasserstein distance, we place an infinite cost on changes in the arrival times of the events. This has the interpretation that our uncertainty is not on the arrival time of the tail events but rather on the dependence. We formalize this through the following cost function between two points:

$$c((s,u),(t,v)) = \infty \mathbb{1}_{s \neq t} + |u - v| \mathbb{1}_{s=t} \qquad (6)$$

Our Wasserstein cost is then between the spectral component while fixing the radial component. We formalize this case in the following corollary.

**Corollary 3.2.** *For the given setting in Theorem 3.1, if the point process satisfies* $N(x) = \lim_n \sum_{i=1}^n \mathbb{1}_{X^{(i)}/n}(x)$, *where $X^{(i)}$ are i.i.d. standard Frechét margins. Then, we can rewrite the primal problem as:*

$$\sup_{W_c(P,P_0) \leq \delta} \mathbb{E}_{\widetilde{N} \sim P} \left[ \ell(\widetilde{N}(f \circ T^{-1})) \right], \qquad (7)$$

*where $\widetilde{N}(da, dv) = \sum_n \mathbb{1}_{(A^{(n)}, Y^{(n)})}(da, dv)$ and $T(X) = (1/\|X\|_1, X/\|X\|_1)$. Under $P_0$, $A^{(n)}$ is the n-th arrival of the Poisson point process with intensity 1 and $Y^{(n)}$ follows the spectral measure $H(\cdot)$. Here we introduce the composition of the Borel function $f$ on the transformation $T$.*

# 4 OPTIMIZATIONS FOR SPECIFIC LOSS FUNCTIONS

Having introduced the different perspectives of the optimization, we provide specific loss functions where each perspective is particularly conducive towards specific robustification cases.

## 4.1 ROBUSTIFYING THE CDF

Consider the cumulative density function (CDF) of $M \in \mathbb{R}^d$ satisfying $M_i = \max_{n=1}^\infty \frac{Y_i^{(n)}}{A^{(n)}}$, where $Y^{(n)} \in \mathbb{R}^d$ are i.i.d. random vectors and $A^{(n)}$ is the $n$-th arrival time of a unit Poisson point process. The CDF of $M$ satisfies:

$$P(M_1 \leq x_1^{-1}, \ldots, M_d \leq x_d^{-1}) =$$
$$\exp \left( -d \int_{\Delta^{d-1}} \max_{i=1}^d x_i w_i H(dw) \right). \quad (8)$$

Here, Equation 8 is derived from Lemma 2.1 by using the max stable decomposition of the random variable, as demonstrated in Equation 1. We will now exploit specific geometric properties of the point process to derive a dual form. Given the definition of $M_i$, we may make a transformation of the CDF algebraically as follows:

$$P(M_1 \leq x_1^{-1}, \ldots, M_d \leq x_d^{-1}) = \qquad (9)$$
$$P \left( \max_{k=1}^d Y_k^{(n)} x_k \leq A^{(n)}, \forall n \right) := P \left( V_x^{(n)} \leq A^{(n)}, \forall n \right).$$

In this case, we can reduce our analysis to this two dimensional space and consider the robustification of the point process on this space, rather than the full $d$ dimensional space of the $Y^{(n)}$'s. A schematic of this representation before robustification is given in Figure 2a. Using this form leads to a dual form given by the following theorem:

**Theorem 4.1** (Formulation for CDF). *Consider the setting in Theorem 3.1 and the loss function $\ell(X) = \mathbb{1}_{X \geq 1}$, the point process $N(\cdot) := \sum_{n=1}^\infty \mathbb{1}_{(A^{(n)}, V_x^{(n)})}(\cdot)$, the function $f(a, v) = \mathbb{1}_{\mathcal{C}}(a, v)$ and $\mathcal{C} = \{(a, v) \in \mathbb{R}^2 \mid a \geq 0, a \leq v\}$. Then, the dual objective is:*

$$\inf_{W_c(P,P_0) \leq \delta} P'(M_1 \leq x_1^{-1}, \ldots, M_d \leq x_d^{-1})$$
$$= 1 - \inf_{\lambda \geq 0} \lambda \delta + \mathbb{E}_{P_0} \left[ 1 - \lambda \min_{n \geq 1} \left( A^{(n)} - V_x^{(n)} \right)^+ \right].$$
$$(10)$$

*Proof.* See Appendix A.2. □

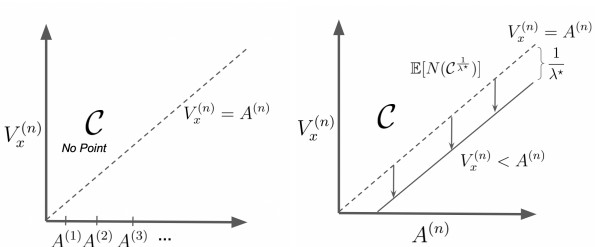

(a) Region of $\mathcal{C}$ before robustification. (b) Shift of region of $\mathcal{C}$ to account for uncertainty.

Figure 2: Two dimensional space for robustificaiton

This formulation is advantageous in many ways. First, we reduced the original optimization problem over measures to one over a scalar variable $\lambda$. Second, this formulation includes the relevant constraints such that the perturbed distribution maintains the max-stability property of EVT. Third, the optimal solution is given by a shift in the rate function, as illustrated in the following corollary:

**Corollary 4.2** (Minimizer of CDF). *The minimizer of the problem in Theorem 4.1 has the following form:*

$$\inf_{P:W_c(P,P_0) \leq \delta} P(X_1 \leq x_1^{-1}, \ldots, X_d \leq x_d^{-1})$$
$$= \exp \left( -\mathbb{E}_{N \sim P_0}[N(\mathcal{C}_{1/\lambda^\star})] \right)$$

where $\mathcal{C}_{1/\lambda^\star}$ is the region $\left\{(a,v) : V_x^{(n)} > A^{(n)} - \frac{1}{\lambda^\star}\right\}$ and $\lambda^\star$ is the minimizer of (10).

*Proof.* See Appendix A.2.1. $\qquad\square$

The ideas behind Corollary 4.2 are illustrated in Figure 2b where the region over which the expectation is taken is shifted by $1/\lambda^\star$.

## 4.2 PROBABILITY AN EVENT IN A RARE SET OCCURS

In this part, we describe a generalization of the CDF case where we are interested in robustifying the probability that an event in the set $A \subset \mathbb{R}_+^d$ occurs, i.e. $P(N(A) \geq 1)$. Our primal problem in this case is:

$$\sup_{W_c(P,P_0)\leq\delta} P(N(A) \geq 1) = \sup_{W_c(P,P_0)\leq\delta} \mathbb{E}_{X\sim P}[\mathbb{1}_{X\in A}]$$

and we let the distance be any $\ell_p$-norm. We can write the dual form and obtain the following simplification:

**Theorem 4.3** (Formulation for Rare Set). *Consider the setting in Theorem 3.1 and the loss function $\ell(X) = \mathbb{1}_{X\geq 1}$, the point process $N(\cdot) = \sum_{n=1}^\infty \mathbb{1}_{X^{(n)}}(\cdot)$, and the function $f(x) = \mathbb{1}_A(x)$. Then the dual objective is:*

$$\sup_{W_c(P,P_0)\leq\delta} P(N(A) \geq 1)$$
$$= \inf_{\lambda\geq 0}\left\{\lambda\delta + \mathbb{E}_{P_0}\left(1 - \lambda\inf_n\inf_{y\in A}\left\|y - X^{(n)}\right\|\right)^+\right\}.$$
(11)

*Proof.* See Appendix A.3. $\qquad\square$

Relatedly, we can also compute the expected number of events to occur within a set using the following theorem:

**Theorem 4.4** (Expected number of events in a set). *Consider the setting in Theorem 3.1 and the loss function $\ell(X) = X$, the point process $N(\cdot) = \sum_{n=1}^\infty \mathbb{1}_{X^{(n)}}(\cdot)$, and the function $f(x) = \mathbb{1}_A(x)$. The the dual objective is:*

$$\sup_{W_c(P,P_0)\leq\delta} \mathbb{E}_{N\sim P}[N(A)]$$
$$= \min_{\lambda\geq 0}\left[\lambda\delta + \mathbb{E}_{P_0}\sum_{n=1}^\infty \left(1 - \lambda\inf_{y\in A}\left\|y - X^{(n)}\right\|\right)^+\right].$$
(12)

## 4.3 CONDITIONAL VALUE AT RISK

Conditional value at risk (CVaR) is a commonly used risk measure in finance. Often there is uncertainty around the calculation, so we derive an efficient form for the calculation.

In the DRO case, we can write the CVaR problem with the given level $\alpha$ as:

$$\sup_{P:W_c(P,P_0)\leq\delta} \inf_z \mathbb{E}_P\left(\frac{(\int \bigvee_{k=1}^d x_k N(dx) - z)^+}{1-\alpha} + z\right).$$
(13)

By exchanging the $\sup - \inf$ to $\inf - \sup$, we can write the dual formulation as the following theorem:

**Theorem 4.5** (Formulation for CVaR). *Consider the setting in Theorem 3.1 and the loss function $\ell(X) = \frac{(X-z)^+}{1-\alpha} + z$, the point process $N(\cdot) = \sum_{n=1}^\infty \mathbb{1}_{X^{(n)}}(\cdot)$, and the function $f(x) = \|x\|_\infty$. The the dual objective is:*

$$\inf_z \sup_{P:W_c(P,P_0)\leq\delta} \mathbb{E}_P\left(\frac{(\int \bigvee_{k=1}^d x_i N(dx) - z)^+}{1-\alpha} + z\right)$$
$$= \frac{\delta}{1-\alpha} + \mathbb{E}_{P_0}\left[\sum_{n=1}^\infty \left\|X^{(n)}\right\|_\infty \,\middle|\, \sum_{n=1}^\infty \|X^{(n)}\|_\infty > q_\alpha\right],$$
(14)

*where $P_0\left(\sum_{n=1}^\infty \|X^{(n)}\|_\infty \leq q_\alpha\right) = \alpha$ and $q_1 < \infty$.*

*Proof.* See Appendix A.5. $\qquad\square$

Having introduced these simplifications, we see that the optimization problems for commonly used losses under the point process interpretation of EVT can be made feasible.

## 5 EXPERIMENTS AND RESULTS

We now consider empirical validation of the proposed optimization schemes using two experiments. The first uses a synthetic data set constructed to pathologically challenge our model and represent real world risk scenarios. The second validation is a baseline comparison using a real data set of financial returns similar to the DRO MEV experiment proposed by Yuen et al. [2020]. For the synthetic and real data experiments the observations are extreme; however, the standard EVD model is shown to not provide appropriate coverage for computing the worst-case risks. For both data sets, we provide empirical results validating that the proposed method satisfies the properties that we desired and outlined in the introduction, i.e. those of sufficient coverage while not being too conservative.

## 5.1 COMPUTATIONAL IMPLEMENTATIONS

For losses that have a reduced dual problem such as those in Section 4, we use the empirical data to represent the expectations and optimize over $\lambda$. However, for general losses we use the equivalence stated in Corollary 3.2 to estimate the adversarial distribution. Our procedure follows

the method in Hasan et al. [2022] where we first estimate the spectral measure from the observations using a generative neural network and then optimize over the space of networks within some ball of the initial distribution. The case for arbitrary $\ell$ is illustrated in Algorithm 1.

---

**Algorithm 1** Procedure for estimating adversarial loss from data.

---

**Require:** Fit $P_0$ according to MEV procedure in Hasan et al. [2022], adversarial budget $\delta$, risk function $\ell(\cdot)$, training iterations $K$.

1: **Initialize:** $P_\theta$, the adversarial distribution as $P_0$
2: **for** $k = 1 \ldots K$ **do**
3:     **Sample:** $Z \sim P_\theta$, $x \sim P_0$
4:     **Compute:** $\mathcal{L}_\lambda(Z, x) = \ell(Z) - \lambda c(Z, x)$
5:     **Solve:** $\max_\theta \mathcal{L}_\lambda(Z, x)$
6:     **Solve:** $\min_{\lambda \geq 0} \lambda \delta + \mathbb{E}_{x \sim P_0} \mathcal{L}_\lambda(Z, x)$
7: **end for**
8: **Return:** Adversarial risk $\lambda^\star \delta + \mathbb{E}\left[\max_Z\left[\ell(Z) - \lambda c(Z, x)\right]\right]$ and adversarial MEV $P_\theta$.

---

## 5.2 EXPERIMENTS WITH SYNTHETIC EVT DATA

We evaluate our proposed methodology on two synthetic two-dimensional EVD data sets.

We hypothesize that a properly formulated DRO approach, where max-stability is enforced via constraints, demonstrates a greater invariance to increased model misspecification relative to a non-constrained models. Our methodology is designed to account for the complexities of real data and produce more accurate and reliable predictions. Details concerning computational parameters, runtime, and code base reference is available in Appendix C.3.

### 5.2.1 Synthetic Datasets

For our synthetic data experiments, we consider mixture distributions where one mixture component with a more concentrated risk appears less frequently than the other mixture component. This is illustrated in Figures 3a and 3b, where the ten thousand observations of the two datasets are visualized. In Figure 3a the component has a smaller tail index and more realizations with smaller magnitude whereas in Figure 3b the component has a larger tail index, but is more rare. For more information see Appendix B.

A standard EVD approach would be sufficient when modeling a non-mixture Symmetric Logistic (SL) or Asymmetric Logistic (ASL) set of data. However, this mixture inserts a pathological modeling issue, which emulates the practical concerns often observed in real data where extrapolating to out-of-sample extreme values is difficult given the scarcity of data in these scenarios. These generated mixture distributions exhibit different tail behavior, which make it difficult

for standard approaches to capture the dependencies accurately, and thus we turn to the proposed DRO estimator and evaluate our ability to recover the true risk.

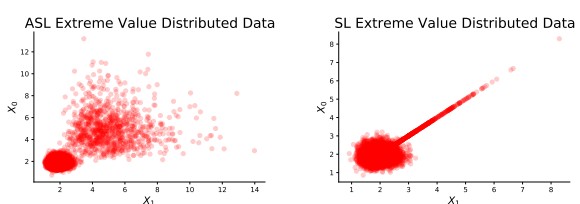

(a) ASL data: lower dependence (b) SL data: higher dependence

Figure 3: Visualization of Synthetic datasets.

### 5.2.2 Synthetic Experiment Results

We now illustrate the results of the computational method for the loss corresponding to CVaR of the $\ell^1$-norm of the observations. Specifically, we aim to robustify the calculation of $\frac{1}{\alpha}\mathbb{E}\left[\|X\|_1 \mathbb{1}_{\|X\|_1 \leq x_\alpha}\right]$ where $x_\alpha = \inf\{x : P(\|X\|_1 \leq x) \geq \alpha\}$. As $\alpha \to 0$, this corresponds to the CVaR for values deep in the tail. We approximate this region as a MEV under both the SL and ASL distributions described previously. Our goal in this experiment is to understand the behavior of the error, defined as the difference between the true risk and the risk computed using the different methods, as a function of the budget $\delta$. We compare 3 different robustifications: (1) a totally unconstrained robustification where $P$ can be in any class of distributions; (2) an EVT constrained case where $P$ must also be an MEV and shares the same $A^{(n)}$'s; (3) an EVT constrained case where the margins are assumed to be well calibrated and only the dependence is modified (such that $w \sim P$, $s.t.$ $\mathbb{E}[w_i] = 1/d$). In terms of standard EVT methods, (3) corresponds to the case where the adversarial distribution has a different dependence structure but maintains the same margins. Specifically, this would suggest estimating the margins is easier and therefore should be conserved, but the dependence structure should be perturbed. Figures 4a and 4b illustrates the error of using a DRO approach with max-stability constraints compared to the unconstrained approach, producing a distribution which

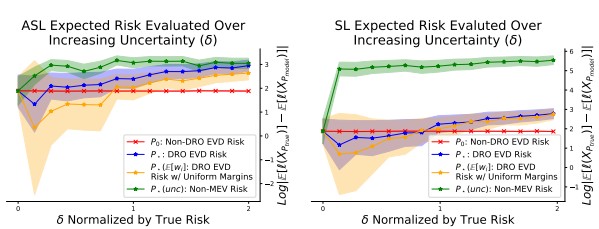

(a) CVaR calculation using lower dependence ASL data (b) CVaR calculation using higher dependence SL data

Figure 4: DRO Estimators for MEV Distributions

is not of the MEV model class as functions of $\delta$. This allows for the understanding of performance based on the behavior of the loss function as $\delta$ changes.

**Synthetic Experiment Results Analysis**   In Figures 4a and 4b, we visualize that the models which are constrained to be max-stable, specifically $P_\star$ and $P_{\star(\mathbb{E}[w_i] \approx \frac{1}{d})}$, demonstrate a lower and robust (positive) error when compared to the unconstrained model $P_{\star(unc)}$. Additionally, we observe a slight improvement in the error of the constrained $P_\star$ model for an increase of model uncertainty ($\delta$) and subsequent monotonically increasing error as uncertainty increases. This behavior confirms our conceptual illustration of the DRO max-stable constrained approach in Figure 1.

### 5.2.3   Convergence of CDF as Distance is Increased

We now consider an experiment that corresponds to Theorem 4.1 described in Section 4.1. We illustrate the behavior in Figure 5 where we see an improvement in the error for a wide range of values of $\delta$.

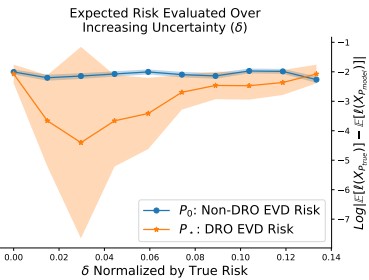

Figure 5: Comparison of adversarial formulations using $A^{(n)}$ and $V^{(n)}$ parameterization of fixed "true value" CDF. As $\delta$ is increased there is greater model uncertainty.

### 5.3   EXPERIMENT ON HIGH DIMENSIONAL FINANCIAL DATA

We consider a dataset on financial returns of equities in the S&P 500 index similar to the observations used in the validation experiments proposed by Yuen et al. [2020]. The dataset is composed of all current members of the S&P 500 who have been members since at least January 1983 to provide forty years of equity price data. We then average returns across companies, where each company is uniquely assigned to one of eleven industrial sectors (industrial, health care, consumer staples, etc.), leading to an eleven dimensional dataset. Maximum daily returns are taken both annually and weekly across the forty year data set, where each of these annual and weekly maximum data sets now constitute extreme observations.

This dataset is in the dimension of the number of industries

considered, in this case $d = 11$. A visualization of this extreme data for highest weekly and annual returns is provided in Appendix B.3. Obtaining a ground truth risk for real data is nearly impossible since we may never observe realizations deep in the tails, so validating against a ground truth risk is difficult. To circumvent this, we consider training on less extreme data given by maxima over shorter time scales and use maxima over longer time scales as the ground truth risk. Specifically, we fit the models using weekly maxima and test on yearly maxima. The reason for this is that large events are more likely to occur over the longer time scale (the year) than over the day, leading to a more representative set of measurements deeper in the tails.

We use the exact same methodology outlined in Section 5.2.2 and illustrate the results of this experiment in Figure 6.

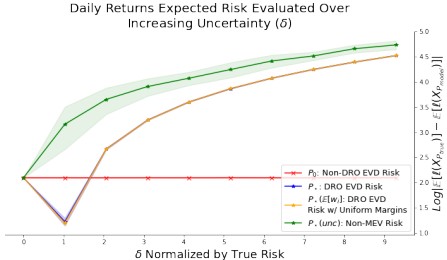

Figure 6: Comparison of different risk estimators on real financial data.

**Real Data Experiment Results Analysis**   We observe that when $\delta =$ true risk, a minimal error is achieved between the true and calculated risk. Additionally, this experiment demonstrates that for a real extreme data set the models constrained by EVT achieve a lower error level when compared to the unconstrained model, as anticipated. We note that the comparable formulation in Yuen et al. [2020] does not provide a modifiable parameter such as our formulation does with $\delta$, to adjust model uncertainty. This demonstrates the additional value of our proposed formulation over a comparable method.

## 6   DISCUSSION

In this paper, we introduced a framework for robustifying MEV distributions according to the point process viewpoint. We provided a novel framework and discussed simplifications that are possible under certain cost functions. We numerically demonstrated that, as we proposed, adding additional constraints from EVT improves the robustification without being too conservative. The framework is evaluated on a synthetic data set which is constructed to challenge the model and a high-dimensional real world data set derived from a similar investigation, demonstrating expected and

improved behavior. The proposed methodology has a variety of applications in many fields, particularly in risk sensitive settings where model misspecification may be present.

There are many avenues for extending the proposed methodology. One important extension involves the case where one must specify a policy to mitigate a worst-case risk. This applies in, for example, portfolio optimization problems where the optimal portfolio minimizes the conditional value at risk over a distribution of portfolio losses. Additionally, further exploration of the proposed formulation leveraging max-stable processes extended to a more general max-infinitely division process may be insightful.

**Limitations** There exist a number of limitations associated with the proposed framework. With regards to the computational method, there may be numerical instabilities associated with the optimization particularly in cases of $\delta$ being small. This is because the $\lambda$ parameter must grow large, which leads to unstable optimization schemes.

### Acknowledgements

Material in this paper is based upon work supported by the Air Force Office of Scientific Research under award number FA9550-20-1-0397.

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

# Distributionally Robust Optimization as a Scalable Framework to Characterize Extreme Value Distributions (Supplementary Material)

**Patrick Kuiper**[*1]   **Ali Hasan**[*1]   **Wenhao Yang**[2]   **Yuting Ng**[1]   **Hoda Bidkhori**[3]   **Jose Blanchet**[2]   **Vahid Tarokh**[1]

[1]Dept. of Electrical and Computer Engineering, Duke University, Durham, North Carolina, USA
[2]Dept. of Management Science and Engineering, Stanford University, Stanford, California, USA
[3]Computational and Data Sciences Dept., George Mason University, Fairfax, Virginia, USA

## A   PROOFS

### A.1   PROOF OF EQUIVALENCE IN REPRESENTATIONS

*Proof.* Let us first recall the two problems that we are solving. On one hand, we take an explicit point process view point in:

$$\max_{W_c(P,P_0)\leq\delta} \mathbb{E}_{N\sim P}\left[\ell(N(f))\right],\qquad\text{(Point Process)}$$

where $N(dx) = \sum_n \delta_{X^{(n)}}(dx)$ is a non-homogeneous Poisson process. By the properties of EVT, the intensity satisfies

$$\mathbb{E}[N(f)] = \int f(T^{-1}(r,w))r^{-2}\mathrm{d}r H(\mathrm{d}w).$$

By change of variables, we could decompose $X^{(n)} = \frac{Y^{(n)}}{A^{(n)}}$, where $A^{(n)}$ is the $n$-th arrival of Poisson process with intensity 1 and $Y^{(n)} \sim H(\cdot)$. Thus, we could rewrite the problem with:

$$\max_{W_c(P,P_0)\leq\delta} \mathbb{E}_{N'\sim P}\left[\ell(N'(f\circ T^{-1})\right],\qquad(15)$$

where $N'(dt,dy) = \sum_n \mathbb{1}_{(A^{(n)},Y^{(n)})}(dt,dy)$ and $T(X) = (1/\|X\|_1, X/\|X\|_1)$.   □

### A.2   CDF PROOF

We now detail the proofs for specific loss functions, beginning with the cumulative distribution function (CDF), followed by the conditional value at risk (CVaR), probability, and expected number of events in rare sets.

*Proof.* Define the upper triangular region as $\mathcal{U} = \{(a,v) : a \geq 0, v \geq a\}$. To compute the CDF, we want to condition on the Poisson process having no points within $\mathcal{U}$, i.e. $N(\mathcal{U}) = 0$. We can compute the rate of the point process according to $\mathbb{E}[N(\mathcal{U})]$. Since this is a Poisson point process, the probability that no points occur within $\mathcal{U}$ is given by $\exp(-\mathbb{E}[V_x])$. Having written the CDF in terms of the rate of the Poisson point process, we now follow through to describe the simplified form written in (10). First, we write the dual problem when directly applying Blanchet and Murthy [2019] as

$$\min_{\lambda\geq 0}\left\{\lambda\delta + \mathbb{E}_{N\sim P_0}\left[\max_{N'}\mathbb{1}_{N'(\mathcal{U})\geq 1} - \lambda\widetilde{c}(N,N')\right]\right\}.$$

---

[*]Authors equally contributed to this work
[*]Authors equally contributed to this work

Conditioning directly on the first inequality, we can rewrite this as

$$\min_{\lambda \geq 0} \left\{ \lambda \delta + \mathbb{E}_{N \sim P_0} \left[ 1 - \lambda \min_{N'(\mathcal{U}) \geq 1} \tilde{c}(N, N') \right]^+ \right\}.$$

Here we also set $c((s, u), (v, t)) = |u - v| \mathbb{1}_{s=t} + \infty \mathbb{1}_{s \neq t}$. Moreover, we notice the equivalent representation between $X^{(n)}$ and $(A^{(n)}, V_n(x))$, which means we can write $N(dx) = \sum_n \mathbb{1}_{X^{(n)}}(dx) = \sum_n \mathbb{1}_{(A^{(n)}, V_n(x))}(dt, dv)$. Thus, if $N(\mathcal{U}) \geq 1$, it means $\exists n$ s.t. $V_n(x) - A_n \geq 0$, we can simply set $N'(\cdot) = N(\cdot)$ to obtain the minimum of $\min_{N'(\mathcal{U}) \geq 1} c(N, N')$. On the other hand, if $N(\mathcal{U}) = 0$, it means $\forall n$, $V_n(x) - A_n < 0$. Then we denote $\delta = \min_n (A^{(n)} - V_n(x))$ and $n^* \in \arg\min_n (A^{(n)} - V_n(x))$ and set $N'(dt, dv) = \mathbb{1}_{(A^{(n^*)}, V_{n^*}(x) + \delta)}(dt, dv) + \sum_{n \neq n^*} \mathbb{1}_{(A^{(n)}, V_n(x))}(dt, dv)$, which guarantees $N'(\mathcal{U}) \geq 1$, and the minimum of $\min_{N'(\mathcal{U}) \geq 1} c(N, N')$ is exactly $\delta$. To conclude, we have the following equivalence holding:

$$\min_{N'(\mathcal{U}) \geq 1} c(N, N') = \min_{n \geq 1} \left[ A^{(n)} - V_x^{(n)} \right]^+.$$

Substituting, we obtain the desired result of (10):

$$\min_{\lambda \geq 0} \lambda \delta + \mathbb{E}_{N \sim P_0} \left[ 1 - \lambda \min_{n \geq 1} \left( A^{(n)} - V_x^{(n)} \right)^+ \right]^+$$

$\square$

### A.2.1  Minimizer of CDF Proof

*Proof.* Following where we ended in the proof of Theorem 4.1, Denote $f(A, V_x) := \min_{n \geq 1} \left( A^{(n)} - V_x^{(n)} \right)^+$. By first order condition of the Equation (10), we have:

$$\delta = \mathbb{E}_{N \sim P_0} \mathbb{1} \left( f(A, V_x) \leq \frac{1}{\lambda^*} \right) f(A, V_x). \tag{16}$$

Substituting it to objective, we have:

$$\min_{\lambda \geq 0} \lambda \delta + \mathbb{E}_{N \sim P_0} \left[ 1 - \lambda \min_{n \geq 1} \left( A^{(n)} - V_x^{(n)} \right)^+ \right]^+$$
$$= P_0 \left( f(A, V_x) \leq \frac{1}{\lambda^*} \right). \tag{17}$$

Thus, we have:

$$\min_{P: W_c(P, P_0) \leq \delta} P(X_1 \leq x_1^{-1}, \ldots, X_d \leq x_d^{-1})$$
$$= 1 - P_0 \left( f(A, V_x) \leq \frac{1}{\lambda^*} \right)$$
$$= P_0 \left( f(A, V_x) > \frac{1}{\lambda^*} \right)$$
$$= P_0 \left( N(\mathcal{C}_{1/\lambda^*}) = 0 \right)$$
$$= \exp \left( -\mathbb{E}_{P_0} N(\mathcal{C}_{1/\lambda^*}) \right) \tag{18}$$

$\square$

## A.3 PROOF OF RARE REGION PROBABILITY

*Proof.* We write the primal problem as

$$\sup_{D(P,P_0)\leq\delta} P(N(A)\geq 1)$$

and writing it in the dual form we get

$$\inf_{\lambda\geq 0}\left\{\lambda\delta + \mathbb{E}_{P_0}\left[\sup_{Z^{(n)}}\left(\sup_{n=1}^{\infty}\mathbb{1}_{Z^{(n)}\in A} - \lambda\sum_{n=1}^{\infty}\|Z^{(n)} - X^{(n)}\|\right)\right]\right\}.$$

Replacing the inner maximization, we can write

$$\sup_{Z^{(n)}}\left(\sup_n \mathbb{1}_{Z^{(n)}\in A} - \lambda\sum_{n=1}^{\infty}\|Z^{(n)} - X^{(n)}\|\right) = \left(1 - \lambda\inf_n\inf_{z\in A}\|z - X^{(n)}\|\right)^+,$$

which is due to:

$$\sup_{Z^{(n)}}\left(\sup_n \mathbb{1}_{Z^{(n)}\in A} - \lambda\sum_{n=1}^{\infty}\|Z^{(n)} - X^{(n)}\|\right) = \left(1 - \lambda\inf_{\exists n,Z^{(n)}\in A}\sum_{n=1}^{\infty}\|Z^{(n)} - X^{(n)}\|\right)^+. \tag{19}$$

And solving $\inf_{\exists n,Z^{(n)}\in A}\sum_{n=1}^{\infty}\|Z^{(n)} - X^{(n)}\|$, we have it equals $\inf_n\inf_{z\in A}\|z - X^{(n)}\|$ as the region only requires one sample $z^{(n)}\in A$ and the others can be equal to corresponding $X^{(n)}$. $\qquad\square$

## A.4 PROOF OF NUMBER OF RARE REGION OCCURRENCES

*Proof.* We begin with the dual definition

$$\inf_{\lambda\geq 0}\left\{\lambda\delta + \mathbb{E}_{N\sim P_0}\left[\sup_{Z^{(1)}\ldots Z^{(N)}}\left(\sum_{n=1}^{\infty}\mathbb{1}_{Z^{(n)}\in A} - \lambda\sum_{n=1}^{\infty}\|Z^{(n)} - X^{(n)}\|\right)\right]\right\}.$$

Rewriting the part within the sup, we obtain the desired result, since

$$\sup_{Z^{(1)}\ldots Z^{(N)}}\left(\sum_{n=1}^{\infty}\mathbb{1}_{Z^{(n)}\in A} - \lambda\sum_{n=1}^{\infty}\|Z^{(n)} - X^{(n)}\|\right) = \sum_{n=1}^{\infty}\left(1 - \lambda\inf_{z\in A}\|z - X^{(n)}\|\right)^+.$$

$$\square$$

## A.5 CVAR PROOF

*Proof.* At first, we consider the finite point case, where $N(\cdot) = \sum_{n=1}^{N}\delta_{X^{(n)}}(\cdot)$. The CVaR problem is given as

$$\min_z \underbrace{\max_{D(P,P_0)\leq\delta}\mathbb{E}_P\left(\frac{\left(\int\bigvee_{i=1}^{d} x_i N(dt,dx) - z\right)^+}{1-\alpha} + z\right)}_{\text{primal problem}}.$$

Taking the primal problem component, we can write the dual problem as:

$$\min_{\lambda\geq 0}\left\{\lambda\delta + \mathbb{E}_{P_0}\left[\max_{Z^{(n)}}\left(\sum_{n=1}^{\infty}\|Z^{(n)}\|_\infty - z\right)^+ - \lambda\sum_{n=1}^{\infty}\|Z^{(n)} - X^{(n)}\|_\infty\right]\right\}$$

$$= \min_{\lambda\geq 1}\left\{\lambda\delta + \mathbb{E}_{P_0}\left(\sum_{n=1}^{N}\|X^{(n)}\|_\infty - z\right)^+\right\} \quad \text{(Lemma A.1)}$$

$$= \delta + \mathbb{E}_{P_0}\left(\sum_{n=1}^{N}\|X^{(n)}\|_\infty - z\right)^+.$$

Adding the minimization over $z$, we obtain

$$\min_z \left[ \frac{\delta + \mathbb{E}_{P_0}\left(\sum_{n=1}^N \|X^{(n)}\|_\infty - z\right)^+}{1-\alpha} + z \right].$$

Taking the minimum, we get

$$\frac{\delta}{1-\alpha} + \mathbb{E}_{P_0}\left(\sum_{n=1}^N \|X^{(n)}\|_\infty \mid \sum_{n=1}^N \|X^{(n)}\|_\infty \geq q_\alpha\right)$$

where $P_0\left(\sum_{n=1}^N \|X^{(n)}\|_\infty \leq q_\alpha\right) = \alpha$. Finally, as $q_1 < \infty$, the summation converges a.s., we can safely let $N \to \infty$ and obtain the final result. $\qquad\square$

**Lemma A.1.** *For $x \in \mathbb{R}_+^d$ and $\lambda \geq 0$, we have:*

$$\max_{y_1\cdots y_N \in \mathbb{R}_+^d} \left(\sum_{n=1}^N \|y_n\|_\infty - z\right)^+ - \lambda \sum_{n=1}^N \|y_n - x_n\|_\infty = \begin{cases} \infty, & \text{when } \lambda \in [0,1), \\ \left(\sum_{n=1}^N \|x_n\|_\infty - z\right)^+, & \text{when } \lambda \in [1,\infty). \end{cases} \quad (20)$$

*Proof.* For $\lambda \in [0,1)$, we claim the objective value would diverge to infinity. In fact, we let $y_n = x_n$ for $n \geq 2$. The objective becomes:

$$\max_{y_1 \in \mathbb{R}_+^d} \left(\|y_1\|_\infty + \sum_{n=2}^N \|x_n\|_\infty - z\right)^+ - \lambda \|y_1\|_\infty. \quad (21)$$

Then, we can pick $y_1$ such that $\|y_1\|_\infty \geq z - \sum_{n=2}^N \|x_n\|_\infty$ and letting it goes infinity, we find the objective will also go infinity.

For $\lambda \in [1,\infty)$, we have:

$$\left(\sum_{n=1}^N \|y_n\|_\infty - z\right)^+ - \lambda \sum_{n=1}^N \|y_n - x_n\|_\infty \leq \left(\sum_{n=1}^N \|y_n - x_n\|_\infty + \|x_n\|_\infty - z\right)^+ - \lambda \sum_{n=1}^N \|y_n - x_n\|_\infty$$

$$= (1-\lambda) \sum_{n=1}^N \|y_n - x_n\|_\infty + \left(\sum_{n=1}^N \|x_n\|_\infty - z\right)^+. \quad (22)$$

Thus, maximizing over $y$, we have the upper bound for the objective is $\left(\sum_{n=1}^N \|x_n\|_\infty - z\right)^+$. And this value is attained when $y_n = x_n$ for $n = 1, \cdots, N$. $\qquad\square$

# B  DATA DESCRIPTION

Here we provide more details on the data distributions used. The symmetric logistic (SL) and asymmetric logistic (ASL) distributions are commonly used parametric models that have the property of max-stability. We describe some of their properties and how they are applied in the mixture model. We use the sampling algorithms described in Stephenson [2003] to generate all the datasets.

As mentioned, for computing the cumulative distribution, the central object of interest is the spectral measure. In each of these datasets, the spectral measure is given by a mixture of two spectral measures. Our baseline model $P_0$ is fit to the mixture without consideration that the model contains two components and results in model misspecification.

## B.1 SYMMETRIC LOGISTIC MIXTURE

For these experiments, we consider a mixture model with two components: one that is almost completely dependent and another that is almost completely independent. The component with dependence is sampled with lower probability than the one with independence. Therefore, a naive estimator would largely concentrate on the component with independence than the one with dependence. The dependence function for this distribution is given by

$$A_{SL}(w) = \left( \sum_{i=1}^{d} w_i^{\frac{1}{\alpha}} \right)^{\alpha}, \quad \alpha \in (0, 1].$$

Notice that the influence $\alpha$ is equal for all components of the vector. We specifically sample the distribution with large dependence ($\alpha = 0.1$) with probability 0.1 whereas we sample the distribution with small dependence ($\alpha = 0.9$) with probability 0.9.

## B.2 ASYMMETRIC LOGISTIC MIXTURE

While the SL distribution is symmetric and all components have the same level of dependence, the ASL can impose specific dependencies between any subset of variables. Denoting $\mathcal{P}_d$ as the power set of $\{1, \ldots, d\}$, the dependence function for this family of distributions is given by

$$A_{ASL}(w) = \sum_{b \in \mathcal{P}_d} \left( \sum_{i \in b} (\lambda_{i,b} w_i)^{\frac{1}{\alpha_b}} \right)^{\alpha_b}, \quad w \in \Delta_{d-1}, \alpha \in (0, 1), \lambda_b \in \Delta_{|b|-1}.$$

We then consider a mixture of two different components, one again with a heavier tail and another with a lighter tail. This is given by setting $\lambda$ to be the same random point on the simplex for both mixture components but then considering $\alpha = 0.1$ and $\alpha = 0.9$ for the components with high dependence and low dependence, respectively.

## B.3 FINANCIAL DATA AND VISUALIZATION

Below we've provided a visualization of the maximum daily returns taken over weekly blocks, and maximum daily returns taken over annual blocks. The returns are provided for the eleven industries, where the values are averages taken across selected companies uniquely belonging to each industry. To compute the average of each industry, we first selected the 93 members of the S&P 500 that have been listed on the index since January 1983, in order to provide forty years of financial data. Each of these company's daily returns were averaged across their assigned industry to create the average daily returns for each industry over forty years.

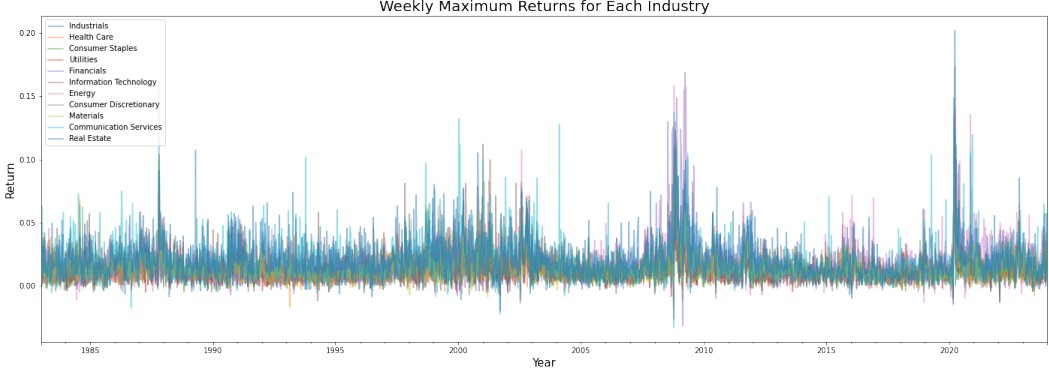

(a) Maximum Weekly Returns

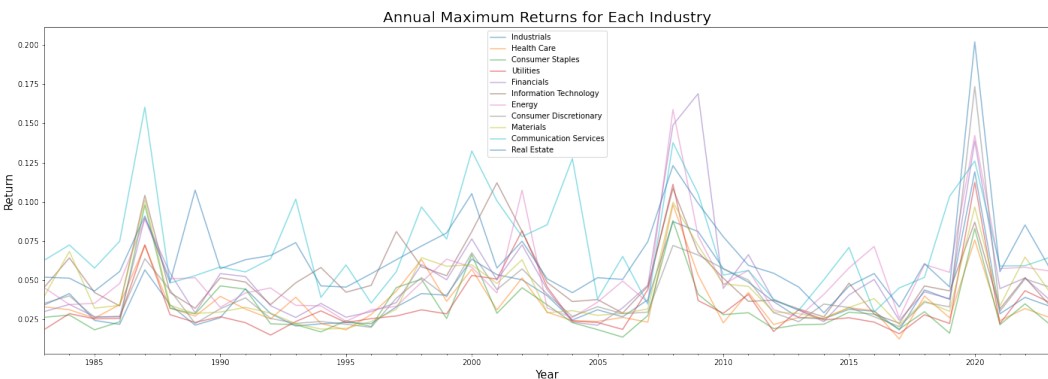

(b) Maximum Annual Returns

Figure 7: Extreme Return data for Eleven Industries

## C ALGORITHM DETAILS

Here we provide additional details that were left out of the main manuscript. Specifically, we go over the **Maximize** and **Minimize** steps in greater detail. The basic idea is we have an iterative training algorithm similar to the $\min - \max$ problems in generative adversarial networks. $\lambda$ is initialized according to $\frac{1}{\delta + \varepsilon}$ for some small $\varepsilon$. $P$ is initialized according to the same parameters as $P_0$. For each of the $K$ iterations, we

**Maximization step:** We fix $\lambda$ to be the value from the last iteration.

1. Sample $N$ points from the base distribution $X^{(n)} \sim P_0, \quad n = 1, \ldots, N$ to obtain the $X$ points from the base distribution.

2. Sample $N$ points from the current estimate of the adversarial distribution $P$ to get $\tilde{X}^{(n)} \sim P, \quad n = 1 \ldots N$.

3. Compute $\mathcal{L}$ for each $X^{(n)}, \tilde{X}^{(n)}$ which results in a $N \times N$ matrix of values, i.e.

$$\mathbf{L} := \begin{pmatrix} \mathcal{L}_\lambda(\tilde{X}^{(1)}, X^{(1)}) & \cdots & \mathcal{L}_\lambda(\tilde{X}^{(1)}, X^{(n)}) \\ \vdots & \ddots & \vdots \\ \mathcal{L}_\lambda(\tilde{X}^{(n)}, X^{(1)}) & \cdots & \mathcal{L}_\lambda(\tilde{X}^{(n)}, X^{(n)}) \end{pmatrix} \tag{23}$$

4. Take the $\max$ over the $\tilde{X}^{(n)}$ dimension, i.e. $\max_i \mathbf{L}_{i,\cdot}$

5. Average over the remaining $Y^{(n)}$ dimension, i.e. $R = \frac{1}{N} \sum_{j=1}^N \max_{i=1}^N \mathbf{L}_{i,j}$.

6. Compute the gradient of $R$ with respect to the parameters of $P$, update the parameters of $P$ via gradient descent.

Adding different constraints to the problem results in different forms of the $\tilde{Y}$, which we describe in the next section.

**Minimization step:** Now we fix the parameters of $P$ and only minimize over $\lambda$.

1. Compute $\hat{R} = \lambda\delta + \frac{1}{N}\sum_{j=1}^{N}\max_{i=1}^{N}\mathbf{L}$.

2. Compute gradient of $\hat{R}$ with respect to $\lambda$, update according to gradient descent.

We next describe a few algorithmic considerations of the maximization step.

## C.1 UNCONSTRAINED CASE

When sampling from an MEV, one samples according to $A^{(n)}, Y^{(n)}, \; n = 1, \ldots, N, N \gg 0$. Then samples are generated according to $\max_{n=1}^{N}\frac{Y^{(n)}}{A^{(n)}}$ (e.g. see Dombry et al. [2016] for additional considerations). In the unconstrained problem, we let the $X^{(i)}$ in (23) be the MEV, i.e. $X^{(i)} = \max_{n=1}^{N}\frac{Y^{(i,n)}}{A^{(i,n)}}$. Then, $P$ does not use any of the factorization of the $Y^{(n)}/A^{(n)}$ and is free to be whatever distribution maximizes the quantity. The problem is unconstrained because it does not consider any of the constraints given by EVT, specifically that the factorization in terms of the radial $A$ and spectral component $Y$ must hold.

## C.2 CONSTRAINED CASE

On the other hand, we consider the explicit factorization in the constrained case. Here we share the $A^{(n)}$ across both distributions of $P, P_0$. Then, $X$ remains as described in the unconstrained case but $\tilde{X}$ is given by $\tilde{X}^{(i)} = \max_{n=1}^{N}\tilde{Y}^{(i,n)}/A^{(i,n)}$ where the $\tilde{Y}^{(i,n)}$ are sampled from the estimated $P$ distribution. Note that the $A^{(n)}$ are the same for the calculation of $X$ and $\tilde{X}$. The constraints then inform the adversarial distribution that it must be max-stable and that the radial component is consistent between both.

## C.3 COMPUTATIONAL DETAILS AND CODE REFERENCE

The code base for the DRO MEV model, written in python, is available at the following github link: `https://github.com/patrick-kuiper/mev_dro`. Below, in Table 1, we provide the run time for each of the synthetic data experiments: Unconstrained, Constrained Extreme Value Distributed (EVD), and Constrained Extreme Value Distributed: with Unit Margin (EVD-SM). We chose to highlight the synthetic data sets as they represent the most challenging experiments computationally. Each of these experiments were run individually on NVIDIA GeForce RTX 3090 GPUs with CUDA Version 12.2. Each experiment was run for 2001 epochs.

Table 1: Experiment time comparison table.

| Data Type | Experiment Type | Run Time (H:M:S) |
|---|---|---|
| Symmetric Logistic | Unconstrained | 0:09:00 |
| Symmetric Logistic | Constrained: EVD | 0:11:21 |
| Symmetric Logistic | Constrained: EVD-SM | 0:12:22 |
| Asymmetric Logistic | Unconstrained | 0:08:56 |
| Asymmetric Logistic | Constrained: EVD | 0:12:19 |
| Asymmetric Logistic | Constrained: EVD-SM | 0:13:22 |