# OpenReview forum: "Distributionally Robust Optimization as a Scalable Framework to Characterize Extreme Value Distributions"
_auai.org/UAI/2024/Conference — UAI 2024 poster_

### Official Review · Reviewer_sS2A · 2024-03-09

**Q2-1 Originality-Novelty:** 3
**Q2-2 Correctness-Technical Quality:** 3
**Q2-5 Clarity Of Writing:** 3

**Q1 Summary And Contributions:**

This paper introduces distributionally robust optimization (DRO) estimators tailored for multidimensional Extreme Value Theory (EVT) statistics, leveraging semi-parametric max-stable distributions derived from spatial Poisson point processes. Recognizing the challenges posed by the scarcity of extreme data and the potential for model misspecification, the study proposes DRO estimators that incorporate max-stable constraints to balance the risk of over-conservatism against out-of-sample performance improvements. The research explores both convex formulations for specific problems, such as Conditional Value at Risk (CVaR), and advanced estimators based on neural networks. Validation on synthetic datasets confirms the effectiveness of these methods, and application to a real financial returns dataset illustrates the model's innovative edge over previous analyses. The work positions the proposed DRO estimators as a significant advancement in applying EVT to model extreme events with enhanced robustness and accuracy.

**Q2-3 Extent To Which Claims Are Supported By Evidence:**

3: Good: the main claims are supported by convincing evidence (in the form of adequate experimental evaluation, proofs, (pseudo-)code, references, assumptions).

**Q2-4 Reproducibility:**

3: Good: key resources (e.g. proofs, code, data) are available and key details (e.g. proofs, experimental setup) are sufficiently well-described for competent researchers to confidently reproduce the main results.

**Q3 Main Strengths:**

1. The paper is easy to follow.

2. The proposed method has theoretical proof to support it.

3. The motivation is clear.

**Q4 Main Weakness:**

1. It seems like the proposed method is too complex. Therefore, it is better to show the time complexity in experiments.

**Q5 Detailed Comments To The Authors:**

N/A

**Q9 Complying With Reviewing Instructions:**

Yes

---

> ### Author Rebuttal · Authors · 2024-04-07
>
> **Adding time complexity.**
> We completely agree with the suggestion to add a discussion on the time associated with the proposed method, and will include a discussion associated with this point in the final draft of the paper. In summary, while the formulation may appear complex, the run time associated with the method is reasonable given the size of the data set. Below, we provide the run time for each of the synthetic data experiments: Unconstrained, Constrained Extreme Value Distributed (EVD), and Constrained Extreme Value Distributed: with Unit Margin (EVD-SM). We chose to highlight the synthetic data sets as they represent the most challenging experiments computationally. Each of these experiments were executed individually on NVIDIA GeForce RTX 3090 GPUs with CUDA Version 12.2. Each experiment was run for 2001 epochs. This information will be added to the final draft of the paper.
>
> **Data Type / Experiment Type / Run Time (H:M:S).**
>
> Symmetric Logistic / Unconstrained / 0:09:00
>
> Symmetric Logistic / Constrained: EVD / 0:11:21
>
> Symmetric Logistic / Constrained: EVD-SM / 0:12:22
>
> Asymmetric Logistic / Unconstrained / 0:08:56
>
> Asymmetric Logistic / Constrained: EVD / 0:12:19
>
> Asymmetric Logistic / Constrained: EVD-SM / 0:13:22

---

### Official Review · Reviewer_YiG3 · 2024-03-19

**Q2-1 Originality-Novelty:** 3
**Q2-2 Correctness-Technical Quality:** 3
**Q2-5 Clarity Of Writing:** 3

**Q1 Summary And Contributions:**

This paper studies the distributionally robust optimization estimators for multidimensional extreme value theory. Motivated by the failures of the current max-stable models in robustifying against scenarios in the tails when the data is sub-asymptotic and the assumptions of EVT are violated, this paper proposes novel distributionally robust optimization estimators informed by semi-parametric max-stable constraints in the space of point processes. Also, this paper provides tractable distributionally robust optimization formulations for several estimators, and introduces a more general neural network based estimator. Experiments show that the MEV-constrained adversaries in proposed estimators manage to provide an appropriate balance of coverage while maintaining underlying structural properties.

**Q2-3 Extent To Which Claims Are Supported By Evidence:**

4: Excellent: all claims are supported by very convincing evidence (in the form of comprehensive experimental evaluation, rigorous mathematical proofs, detailed (pseudo-)code, precise references, well-motivated and realistic assumptions) and the authors deliver what they promise.

**Q2-4 Reproducibility:**

3: Good: key resources (e.g. proofs, code, data) are available and key details (e.g. proofs, experimental setup) are sufficiently well-described for competent researchers to confidently reproduce the main results.

**Q3 Main Strengths:**

1. The paper is well-written and the main idea is quite novel.
2. The DRO framework designed in this paper manages to mitigate the main drawbacks of the current max-stable models.
3. The paper provides tractable formulations for several important estimators, which reduced the difficulty of optimization.
4. The proposed estimators achieve good empirical performance compared to the previous work.

**Q4 Main Weakness:**

The paper is well-written,  the following potential advice may make the paper better, and since I'm not an expert on the EVT, if there're technically mistakes in the following advice, I apologize for them and please just ignore them :)

1. If how the equations in Section 3 and 4 are derived from the previous analyses is described more clearly, then the paper will be more readable. For example:


 (1). how we obtained Equation (8) from Lemma 2.1;



 (2). how the Corollary 3.2 is obtained (why we introduce the specific T in Equation (7)?);



 I think if some sentences are added to help the readers understand the process of obtaining these formulations and equations, then the paper will be much more friendly to some readers who're not quite familiar with EVT field.


2. Several notations are not very clear in the paper, and also there're several typos.

(1).In Equation (4), what does "N($\cdot$) $\neq$ N'($\cdot$)" mean? Is it supposed to be "N($S$) $\neq$ N'($S$)"? And what does the support of $N'(\cdot)$ mean? I can understand that we want to define a permutation to map the index of valid x to the index of valid y, but it's kind of confusing when I read that "on the support of $N'(\cdot) $: {$1,2,\cdots,N'(S)$ }  "


(2).In Theorem 3.1, should it be $\tilde{c}(N,N')$ instead of $c(N,N')$?


(3).In Equation (7), what does the P' refer to? Should it be P?


(4). Above Equation (8), "The CDF of X satisfies" should be "The CDF of M satisfies".


(5). In Corollary 4.2, what does  $E_{N \sim P_0} [C_{1/\lambda^*}]$ mean? Does it mean $E_{N \sim P_0} [1_{C_{1/\lambda^*}}]$?

**Q5 Detailed Comments To The Authors:**

Please see the weaknesses section.

**Q9 Complying With Reviewing Instructions:**

Yes

---

> ### Author Rebuttal · Authors · 2024-04-07
>
> **If the Equations in Section 3 and 4 are derived from the previous sections more clearly, then the paper will be more readable.**
> Thank you for this comment and we agree that we should provide a more clear development of the preliminary notation and relationships in Section 2, to support Sections 3 and 4. We will update the paper to reflect this comment, and address all enumerated concerns below:
>
> **Equation (8) from Lemma 2.1.**
> Thank you for this comment and we will update the paper to be more clear. Specifically, Equation (8) is derived from Lemma 2.1 by using the max stable decomposition of the random variable, as demonstrated in Equation (1), when considering a unit rate Poisson point process.
>
> **How is Corollary 3.2 obtained.**
> Thank you for this comment and we will update the paper to be more clear. Specifically, Corollary 3.2 is obtained by introducing the composition of the Borel function $f$ on the transformation $T$. We introduce $T$ in Equation (7) as we transform the point process to a Poisson  process, which is in Appendix A.1
>
> **Several notations are not very clear in the paper, and also there are several typos.**
> Thank you for your careful review of the paper and identification of points which are unclear and typos. We will update the paper to reflect this comment, and address all enumerated concerns below:
>
> **Equation (4).**
> Thank you for highlighting these typos. That is correct, in Equation (4), $N(\cdot) \neq N'(\cdot)$ should be denoted as $N(\mathcal{S}) \neq N'(\mathcal{S})$.
> Additionally, the definition of $N(\cdot)$ is $N(\cdot)=\sum_{i=1}^{\infty}\mathbf{1}_{x_i}(\cdot)$. Thus, the support of $N(\cdot)$ is { $x_1,x_2,\cdots  $ }  $ \cap S$
>
> **Theorem 3.1.**
> Thank you for highlighting this typo. That is correct, in Theroem 3.1, $c(N,N')$ should be denoted as $\tilde{c}(N,N')$. We will correct his typo in the final version.
>
> **Equation (7).**
> Thank you for highlighting this typo. That is correct, in Equation (7), $P'$ should be denoted as $P$. We will correct his typo in the final version.
>
> **Above Equation (8).**
> Thank you for highlighting this typo. It should be $M$ instead of $X$.
>
> **Corollary 4.2.**
> Thank you for highlighting this typo. Indeed, referencing Figure 2a,  $E_{N \sim P}[\mathcal{C}_{1/\lambda^{\star}}]$
>
> should be denoted as $E_{N \sim P}[N(\mathcal{C}_{1/\lambda^{\star}})]$. We will correct this typo in the final version.

---

### Official Review · Reviewer_yhth · 2024-03-22

**Q2-1 Originality-Novelty:** 2
**Q2-2 Correctness-Technical Quality:** 2
**Q2-5 Clarity Of Writing:** 3

**Q1 Summary And Contributions:**

The paper develops distributionally robust optimization estimator for multidimensional Extreme Value Theory. The motivation is to robustified the inferential tasks involving tails to model misspecification.
The paper considers Wasserstein distance to qualify model misspecification. The first approach proposed in the paper is based on the interpretation an MEV Through a point process. From this interpretation theorem 3.1 is proposed it is a application of Blachet and Murthy in the specific context of Point Processes.  The cost in the Wassserstein distance is specified to model uncertainty on the dependence.
In section 4 two robustifications are proposed one on the CDF . The theorem 3.1 is apply to the specific case. The advantage of this approach is to transform the optimization problem over measures to a scalar variable.   Then a generalization based to on the application of theorem 3.1 which focus on the robustification of the that an event in a given set occurs is proposed. Finally, based on the CVaR formulation for the DRO case with Poisson point process and using theorem 3.1 a dual for CVaR is proposed.

The experiment, show that taking into account the constraint of EVT limit the over conservatism of DRO. For some normalized value of \deltat the DRO outperform the classical approach. It is showed that the results on experiments proposed by Yuen et al is equivalent to their results.

**Q2-3 Extent To Which Claims Are Supported By Evidence:**

2: Fair: the main claims are somewhat supported by evidence (but the experimental evaluation may be weak, or does not match entirely with the claims, important baselines may be missing, proofs contain important ideas but lack rigor, algorithmic details are only discussed superficially, references are imprecise, assumptions are not sufficiently motivated or explicated, etc.).

**Q2-4 Reproducibility:**

2: Fair: key resources (e.g. proofs, code, data) are unavailable but key details (e.g. proof sketches, experimental setup) are sufficiently well-described for an expert to confidently reproduce the main results.

**Q3 Main Strengths:**

The paper is clear. The experimentations show the interest of taking into account the specificity of the constraint on the distribution form EVT.

**Q4 Main Weakness:**

Have similar results than Yuen et al 2022.
The parameter \deltat is very sensitive and may lead to worst results than without DRO approach. This point is not

**Q5 Detailed Comments To The Authors:**

Sensitivity to \deltat, the fact that the DRO approach can be worse than the non-DRO approach if \deltat is too high, and how to choose a \deltat should be discussed further.
Minor:
Figure 1 could you specify what means uncertainty size in our context.
(7) ~P’=> ~P
For me, the transformation in (9) require few details
Algorithm 1, ligne 5 Why it is a max_{\theta} it is max_{Z}? ligne 6 is not clear for me.

**Q9 Complying With Reviewing Instructions:**

Yes

---

> ### Author Rebuttal · Authors · 2024-04-07
>
> **Similar results to Yuen et al 2022.**
> Thank you, while Yuen et. al. (Yuen) and our proposed model seek to robustify multivariate extreme value estimators, the tools used are different and outcome of our proposal is more general. Yuen focuses on applying constraints to extremal coefficients to solve extreme VAR problems - specifically applicable to financial and insurance markets. Yuen formulates a convex optimization program to solve for upper and lower bonds of extreme VAR.  This is a limited subset of application problems when compared to our investigation. We cover general risk measures using generative modeling, providing closed form solutions for commonly used measures such as CVaR (point process formulation) and the CDF. We achieve these more general DRO estimators by taking advantage of the DRO formulation afforded by the wasserstein metric, reformulating the primal DRO into a dual. We solve this dual using a neural network. We agree and will highlight the novelty of our method compared to Yuen.
>
>
> **Sensitivity to delta t.**
> We agree that we should discuss more clearly the DRO results' sensitivity to the $\delta$ parameter, specifically why large values of $\delta$ may lead to worse results than without the DRO approach. It is true that for values where $\delta$ is large, the DRO EVD under performs the baseline non-DRO EVD estimator; however, this behavior is to be expected. This is because the large values of $\delta$, given the adversarial nature of the DRO program, encourages the formulation to seek out distributions which are far from the closest EVD. Our proposed method is useful in the sense that it provides practitioners with a means to calibrate $\delta$ to lower error, and this fact is demonstrated in our practical application of the proposed methodology in Section 5.3 Experiment on High Dimensional Financial Data. A major focus of the performance comparison is between DRO characterizations which are max-stable to those which are not. Here again we see the proposed max stable risk estimators are robust, showing the lowest error.
>
> **Figure 1 Uncertainty Size Context.**
>  In Figure 1, uncertainty size describes a confidence in the data used for extrapolating EVT distributions, usually quantified by the amount of data available for this process.
>
> **In Equation (7) $P’$ should be $P$.**
> Thank you for pointing out this typo. Yes, in Equation (7) the adversarial distribution $~P’$ should be $P$. We will update this error in the final submission.
>
> **The transformation in Equation (9) requires more details.**
> Thank you for this comment and we agree Equation (9) requires more details. Please note we decompose the random variable $M$ by $M_i=\max_{n=1}^{\infty}\frac{Y_i^{(n)}}{A^{(n)}}$ for each $i$. Then, transformation $(9)$ is induced directly from this by algebra. We will provide this additional discussion in the final draft.
>
> **Algorithm 1, line 5 $max_{\theta}$ vs $max_{Z}$?**
> Thank you for this comment. In line 5 of Algorithm 1, there is a $\max_{\theta}$ because this optimization is generating adversarial data leveraging a neural network ($NN_{\theta}$), where $\theta$ are the parameters of this network: $ Z \sim NN_{\theta}$. We will update Algorithm 1 to be more clear in the final draft.
>
> **Algorithm 1, line 6 is not clear for me.**
> Thank you for pointing out a typo error in Algorithm 1, line 6.  We will clarify  this issue by adding that we are optimizing over the dual parameter $\lambda \geq 0$. We will make several other changes to Algorithm 1 to further clarify the method.

---

### Official Review · Reviewer_VHAT · 2024-03-24

**Q2-1 Originality-Novelty:** 3
**Q2-2 Correctness-Technical Quality:** 3
**Q2-5 Clarity Of Writing:** 4

**Q10 Ethical Concerns:**

No ethical issues were found.

**Q1 Summary And Contributions:**

This paper introduces novel distributionally robust optimization (DRO) estimators for multidimensional Extreme Value Theory (EVT), aiming to address model misspecification due to the rarity of extreme data. It explores both convex and neural network-based approaches, demonstrating reliability and out-of-sample performance improvements. The methods were validated on synthetic and real financial data, establishing the proposed model as a new solution in the EVT domain.

**Q2-3 Extent To Which Claims Are Supported By Evidence:**

3: Good: the main claims are supported by convincing evidence (in the form of adequate experimental evaluation, proofs, (pseudo-)code, references, assumptions).

**Q2-4 Reproducibility:**

3: Good: key resources (e.g. proofs, code, data) are available and key details (e.g. proofs, experimental setup) are sufficiently well-described for competent researchers to confidently reproduce the main results.

**Q3 Main Strengths:**

Overall, the paper’s innovative approach, comprehensive methodology, and successful empirical validation highlight its strengths and its potential impact on the field of Extreme Value Theory and its applications.

**Q4 Main Weakness:**

The main weaknesses lie in the lack of discussion on potential failure scenarios of the proposed method and its scalability.

**Q5 Detailed Comments To The Authors:**

Add more references for non-experts in this field to the claims mentioned in the introduction. For example, at the end of the first paragraph of the introduction.

**Q9 Complying With Reviewing Instructions:**

Yes

---

> ### Author Rebuttal · Authors · 2024-04-07
>
> **Failure Cases.**
> We agree with the comment that our paper will benefit from a discussion on potential failure scenarios of the proposed method and its scalability. We will add specific discussion of possible failure scenarios of the method. Specifically, we may highlight the case where the parameter associated with the allowable Wasserstein distance, $\delta$, is close to zero. In this case the optimization can be unstable and sometimes the DRO produces poor risk estimations.
>
> **References.**
> We agree with your comment and we will address the addition of references for non-experts in the field of EVT to support the claims mentioned in the introduction. We will accomplish this by further citing and explaining fundamental EVT concepts from well developed resources (such as Stuart Coles' book in the references), which develop in detail both max-stable processes and the use of point processes to described multivariate extreme value distributions.
>
> **Scalability**.
> With respect to scalability specifically, we refer to the run times provided below. We have provide the run time for each of the synthetic data experiments: Unconstrained, Constrained Extreme Value Distributed (EVD), and Constrained Extreme Value Distributed: with Unit Margin (EVD-SM). We chose to highlight the synthetic data sets as they represent the most challenging experiments computationally. Each of these experiments were executed individually on NVIDIA GeForce RTX 3090 GPUs with CUDA Version 12.2. Each experiment was run for 2001 epochs. This information will be added to the final draft of the paper.
>
> **Data Type / Experiment Type / Run Time (H:M:S).**
>
> Symmetric Logistic / Unconstrained / 0:09:00
>
> Symmetric Logistic / Constrained: EVD / 0:11:21
>
> Symmetric Logistic / Constrained: EVD-SM / 0:12:22
>
> Asymmetric Logistic / Unconstrained / 0:08:56
>
> Asymmetric Logistic / Constrained: EVD / 0:12:19
>
> Asymmetric Logistic / Constrained: EVD-SM / 0:13:22

---

### Meta-Review · Area_Chair_U1GG · 2024-04-19

The authors introduce a distributionally robust optimization (DRO) estimators for multidimensional Extreme Value Theory (EVT). By exploiting the duality formulation, the authors obtain the optimization, which is neural network compatible.

Most of the reviewers are positive to this submission.

The major issues raised by the reviewers lie in

- The similarity w.r.t. Yuen et.al., 2020 should be carefully discussed to emphasize the significancy.

- Some notations should be clarified.

Please consider the reviewers' suggestion to improve the submission.